# SERES: La Paz Empieza en Casa—Evaluation of an Intervention Program to Reduce Corporal Punishment and Parenting Stress, and to Enhance Positive Parenting Among Colombian Parents

**DOI:** 10.3390/ejihpe15110223

**Published:** 2025-10-29

**Authors:** Angela Trujillo, Martha Rocío González, José David Amorocho

**Affiliations:** Faculty of Behavioral Sciences, Universidad de La Sabana, Cundinamarca 111321, Colombia; angela.trujillo@unisabana.edu.co (A.T.); joseammo@unisabana.edu.co (J.D.A.)

**Keywords:** parenting intervention, corporal punishment, parental stress, artificial neural networks, positive parenting practices

## Abstract

Background: Corporal punishment (CP) remains a common disciplinary practice in many countries, despite evidence of its negative consequences for children’s development. Objective: This study examined the effectiveness of a culturally adapted intervention aimed at reducing parents’ use of CP. Method: Using a 12-month quasi-experimental longitudinal design, the study included an intervention group (n = 21) and a control group (n = 17). We administered standardized instruments at pretest and posttest to assess changes in parenting behavior, emotional regulation, and perceptions of child behavior. Artificial neural networks (ANNs) were used to model nonlinear relationships and classify group membership. Results: The intervention group showed significant improvements in parenting practices and emotion regulation. The ANN model classified participants with 74.6% accuracy. Key predictive variables included emotional suppression, physical punishment, and parental support and acceptance. Conclusions: These findings provide evidence for the effectiveness of the SERES program in reducing harmful parenting behaviors and promoting positive practices. Additionally, the use of AI models proved to be valuable for understanding complex behavioral changes, offering a promising approach for optimizing future interventions aimed at strengthening parenting and preventing family violence.

## 1. Introduction

Positive parenting practices are essential to the healthy development, well-being, and psychological adjustment of children and adolescents ([31]; [63]). In contrast, parenting strategies rooted in violence—such as physical punishment, harsh treatment, and neglect—are consistently associated with adverse outcomes in both the short and long term ([34]; [41]; [47]; [78]). These harmful practices negatively affect children’s emotional, ([24]) cognitive ([46]), and social development ([3]).

Violence against children constitutes a global public health concern, with particularly high prevalence rates in low- and middle-income countries. It is estimated that approximately six out of ten children aged 2 to 14 experience violent parenting practices from primary caregivers ([70]; [73]). Beyond severely impairing child development, this issue generates considerable economic and social costs ([73]). Consequently, preventing violence against children is increasingly recognized as a critical public health priority in these regions.

### 1.1. Corporal Punishment and Its Associated Effects

Corporal punishment (CP) is defined as the use of force intended to cause pain, but not injury, to correct or control a child’s behavior ([65]). Examples include hitting, spanking, and other forms of physical discipline enforced through the fear of pain. Although CP is generally intended to modify behavior rather than inflict harm, extensive research demonstrates that it is an ineffective disciplinary strategy ([6]; [23]; [24]; [65]). The adverse outcomes associated with CP have been consistently observed across measurement tools, informant sources, and culturally diverse settings, as shown by both cross-sectional and longitudinal studies ([24]).

Exposure to CP has been linked to difficulties in emotion regulation, increasing the likelihood of impulsive and aggressive or avoidant responses in challenging situations ([8]; [27]; [39]; [44]). This impulsivity has been associated with heightened vulnerability to antisocial and delinquent behaviors ([53]).

Moreover, children exposed to CP face elevated risks of anxiety and depression resulting from continual exposure to stressful, fear-inducing environments ([24]). Chronic exposure also detrimentally affects cognitive functions—such as memory and attention—potentially undermining academic success ([11]; [24]). These children may develop negative attitudes toward school, reflected in academic disengagement and diminished motivation ([57]). Over time, CP exposure has been linked to a heightened risk of mental health disorders, including anxiety, depression, and post-traumatic stress disorder ([22]).

Notably, relational patterns learned through childhood experiences of CP often persist into adulthood, manifesting as interpersonal difficulties in romantic, occupational, and familial domains ([16]; [21]). Externalizing behaviors such as aggression and violence contribute to cycles of violence across generations ([10]; [27]; [38]; [44]). Given its association with numerous negative developmental outcomes, CP is considered a harmful practice that violates children’s fundamental rights ([68]).

Despite extensive scientific evidence that has established a strong association between physical punishment and adverse developmental outcomes, it remains prevalent in many cultural contexts ([18]; [71]). Concerningly, a study by [69] ([69]) found that 77% of Colombian parents reported using physical punishment within the past year, signaling a critical need for targeted intervention.

### 1.2. Parenting Stress and Emotion Regulation: Implications for Parenting Practices

Understanding why parents resort to violent disciplinary practices requires an examination of parenting stress. Elevated parenting stress is a significant predictor of harsh and coercive parenting behaviors ([20]; [48]; [67]; [60]). Parenting stress is defined as the distress experienced when parents perceive that the demands of caregiving exceed their coping resources ([17]; [50]).

High stress impairs parental self-regulation, increases emotional reactivity, and reduces patience and empathy, thereby elevating the risk of physical punishment or verbal aggression as disciplinary strategies ([25]). This relationship is particularly salient in contexts marked by socioeconomic adversity, limited access to social support, and exposure to intergenerational violence—factors that exacerbate parenting stress and constrain caregivers’ capacity to engage in positive parenting practices ([37]).

Emotion regulation, conceptualized through [28]’s ([28]) model, encompasses the processes by which individuals influence the emotions they experience, their timing, and their expression ([43]). The model delineates five regulatory strategies: situation selection, situation modification, attentional deployment, cognitive change, and response modulation. These strategies operate along the emotion timeline and are well supported by theoretical and empirical research ([4]). Emotion regulation plays a critical role in parenting, especially under stress. Interventions that address parenting stress—including parent training, emotion regulation support, and community psychosocial resources—show promise in reducing violent parenting and promoting nurturing caregiving ([12]).

### 1.3. Positive Parenting Practices

Parental monitoring involves parents’ knowledge and supervision of children’s daily activities and peer groups. This construct reflects both parental efforts to seek information and children’s voluntary disclosure ([64]). Effective monitoring is linked to reduced adolescent engagement in risky behaviors such as substance use, aggression, and early sexual activity ([19]; [29]; [62]). It also strengthens parent–child communication and emotional support, which are essential for psychosocial development ([55]). However, practitioners often report a lack of structured tools to guide parents in monitoring, underscoring the need for evidence-based interventions ([54]; [62]).

Parental support and acceptance are critical to children’s emotional and psychological development. These elements strengthen children’s sense of self-worth and enhance resilience in the face of adversity ([40]). Support and acceptance refer to parental behaviors and attitudes that demonstrate affection, understanding, respect, and unconditional acceptance toward their children. These behaviors foster an emotionally secure and nurturing environment, promoting socioemotional development, self-esteem, and psychological well-being in both children and adolescents. Empirical evidence indicates that these practices are associated with lower levels of behavioral and emotional problems, as well as enhanced social skills and resilience ([58]).

Parental inductive discipline is a parenting strategy that emphasizes reasoning, explanation, and perspective-taking to guide children’s behavior, in contrast to coercive or punitive methods. This approach has been shown to foster favorable emotional and social outcomes in children, alongside the development of prosocial behaviors, with child inhibitory control and sympathy posited as key mediating mechanisms ([77]). The prevalence of inductive discipline typically increases as children transition from early to middle childhood. During this developmental phase, parents—particularly mothers—tend to shift away from harsher disciplinary strategies, adopting more explanatory and empathetic approaches ([72]). This transition is strongly influenced by children’s cognitive and social maturation, particularly advancements in executive functioning and inhibitory control ([72]). Inductive discipline has been consistently associated with reductions in externalizing behaviors, such as aggression and defiance, and increases in prosocial behavior ([14]; [76]). Its effectiveness is further enhanced when applied alongside parental warmth and adaptive emotion regulation strategies, such as cognitive reappraisal ([76]). Although widely regarded as a constructive and developmentally appropriate strategy, the effectiveness of inductive discipline may vary across cultural contexts and family systems, underscoring the importance of culturally sensitive and contextually responsive approaches in parenting interventions.

### 1.4. Evidence-Based Programs for Reducing Violent Parenting Behaviors

Although the literature acknowledges the importance of evidence-based interventions, many programs lack rigorous evaluations. The World Health Organization ([75]) promotes the use of evidence-based strategies to eliminate violence against children. One key strategy is to support parents and caregivers, who are often the primary perpetrators of physical and emotional violence against children. As a result, parenting interventions are increasingly being implemented at scale, with growing interest from public policymakers.

[49] ([49]) analyzed intervention and prevention programs addressing parental violence against children and concluded that there is limited evidence regarding both intervention and prevention programs that have been rigorously evaluated for effectiveness. These findings underscore the need to direct scientific research toward evaluating programs currently being implemented in the field.

Other reviews have found that parenting-improvement programs are generally structured and predominantly implemented in high-income countries (91%) ([5]). All studies that assessed parenting practices reported improvements following the intervention. The authors conclude that parenting education programs represent an effective strategy for the universal prevention of child violence and maltreatment. These findings underscore the importance of implementing and rigorously evaluating such programs in low- and middle-income countries ([5]; [73]).

Most meta-analyses have focused on the short-term effects of parental interventions. The evaluation of long-term outcomes remains limited, largely due to ethical and financial challenges—such as reliance on waitlist control groups and insufficient resources for extended follow-ups. Nonetheless, because the core aim of parental interventions is to achieve lasting improvements in parenting behaviors, their long-term effectiveness, as examined through longitudinal research, represents the most meaningful and policy-relevant outcome ([7]).

In sum, parental interventions can reduce physical and emotional violence by parents and caregivers over time. Given the growing policy interest and the expansion of these interventions globally, more research is urgently needed to evaluate and validate contextually adapted programs that can sustain long-term effects.

In Colombia, research on targeted programs to reduce parental violence remains sparse despite governmental efforts, revealing a critical gap. This study responds to that need by evaluating the SERES program, designed to reduce violent parental practices and parenting stress while fostering positive parenting behaviors, including emotional support, inductive discipline, and monitoring.

### 1.5. Hypothesis

Changes from pre-test (T1) to post-test (T2) in physical punishment prevalence, parental stress, emotional regulation, and positive parenting practices (monitoring, inductive discipline, and support and acceptance) will enable above-chance prediction of each participant’s group membership (control or experimental).

## 2. Materials and Methods

### 2.1. Design

The study employed a quasi-experimental design with pretest and posttest measurements. Participants (N = 38) were assigned to either an intervention group (n = 21) or a control group (n = 17) ([30]).

According to the study design (Figure 1), pretest measures were administered to all participants to assess the prevalence of physical punishment, levels of parenting stress, self-regulation, and positive parenting practices, including monitoring, inductive discipline, and support and acceptance. Participants were assigned to either the intervention or control group. The intervention group completed the SERES program over a 12-month period, while the control group did not receive any intervention. Following the intervention period, posttest assessments were conducted with parents in both groups.

### 2.2. Participants and Procedure

A total of 38 caregivers participated in the study: 21 in the intervention group and 17 in the control group. The mean age was 34.64 years (SD = 9.92).

Gender: Most caregivers were female—76.5% (n = 13) in the control group and 75.0% (n = 16) in the intervention group; the remainder were male—23.5% (n = 4) control and 25.0% (n = 5) intervention.

Kinship: The primary caregiver role was parent—94.1% (n = 16) control and 85.0% (n = 18) intervention.

Socioeconomic status: Middle class was most frequent—76.5% (n = 13) control and 60.0% (n = 13) intervention—followed by lower class—23.5% (n = 4) control and 35.0% (n = 7) intervention—and upper class—5.0% (n = 1) intervention.

Marital status: Consensual union was most common—70.6% (n = 12) control and 45.0% (n = 10) intervention—followed by married—11.8% (n = 2) control and 20.0% (n = 4) intervention; single—17.6% (n = 3) control and 15.0% (n = 3) intervention; divorced—10.0% (n = 2) intervention; widowed—5.0% (n = 1) intervention; and separated—5.0% (n = 1) intervention.

Education: Secondary education was most prevalent—52.9% (n = 9) control and 45.0% (n = 10) intervention. Other levels included technical/technological—11.8% (n = 2) control and 25.0% (n = 5) intervention; undergraduate—17.6% (n = 3) control; basic primary (ninth grade)—11.8% (n = 2) control; primary—5.9% (n = 1) control and 20.0% (n = 4) intervention; and postgraduate—5.0% (n = 1) intervention.

Participants were recruited in coordination with the Municipal Secretariat of Education in Cundinamarca, Colombia. The research protocol was introduced to interested public educational institutions. During institution-led informational sessions, parents, legal guardians, and other primary caregivers were invited to enroll. Those who agreed provided written informed consent, confirming voluntary participation and confidentiality protections. An initial home visit was then scheduled in collaboration with designated program facilitators. Finally, participants were assigned to either the intervention or control group.

### 2.3. Measures

The Spanish version of the Parent–Child Conflict Tactics Scale (CTSPC; [66]) assesses three dimensions of caregiver behavior: Nonviolent Discipline, Psychological Aggression, and Physical Aggression. In this study, we used the Physical Aggression subscale, which captures behaviors ranging from socially sanctioned corporal punishment to criminal acts of physical assault. Items also measure the frequency with which these practices were employed over the past year, with higher scores indicating more frequent use of physically aggressive practices.

Parental emotion regulation was measured with the revised Parental Emotion Regulation Inventory (PERI-2; [35]), a self-report instrument grounded in [28]’s ([28]) model. The original PERI includes 13 items assessing the extent to which parents use cognitive reappraisal and expressive suppression during discipline encounters. The revised PERI-2 expands coverage to four strategies: suppression (concealing negative emotion), capitulation (yielding to the child’s aversive behavior to reduce one’s own discomfort), escape (temporarily leaving the discipline interaction to relieve distress), and reappraisal (cognitively reframing the child’s behavior to decrease negative affect). Validation work, conducted primarily with parents of toddlers (ages 1–3), supports a four-factor structure (suppression, capitulation, escape, reappraisal) and demonstrates strong internal consistency (e.g., α ≈ 0.74–0.92). Suppression, capitulation, and escape show meaningful associations with harsh parenting, lax discipline, and child aggression, whereas reappraisal has exhibited weaker validity in original samples. Test–retest reliability has not yet been established.

Parenting stress was assessed using the Parenting Stress Index–Short Form (PSI/SF; [2]). Participants rated statements such as “I feel trapped by my responsibilities as a parent” on a 5-point Likert scale ranging from 1 (strongly disagree) to 5 (strongly agree), with higher scores reflecting greater parenting stress. Internal consistency was adequate in prior work (α = 0.87 for fathers and α = 0.86 for mothers).

Parental monitoring was measured using a 9-item scale indexing parents’ knowledge of their children’s behaviors and activities ([36]). Items such as “My parents know who my friends are,” with a corresponding parent version, were rated on a 5-point Likert scale from 1 (“does not know”) to 5 (“knows a lot”). Reliability was high in reference samples (α = 0.91 for fathers, α = 0.90 for mothers, and α = 0.88 for adolescents), and higher scores indicate greater monitoring.

Inductive discipline was evaluated with an 8-item questionnaire developed by [9] ([9]) to assess the extent to which parents use reasoning to draw adolescents’ attention to the consequences of their actions for others. Respondents indicated frequency on a 5-point Likert scale ranging from 1 (“never”) to 5 (“always”). An illustrative item is, “When I set rules for my child, I explain why.” The scale demonstrated strong internal consistency (Cronbach’s α = 0.88), with higher scores reflecting more frequent use of inductive reasoning.

Parental support and acceptance were assessed using the short form of the Parental Acceptance–Rejection Questionnaire ([59]). This 17-item measure captures parental behaviors that convey affection, approval, and appreciation toward the child. Items are rated on a 5-point Likert scale from 1 (strongly disagree) to 5 (strongly agree), with higher scores indicating greater support and acceptance. Internal reliability was good in prior studies (α = 0.86).

Finally, a researcher-designed sociodemographic questionnaire collected information about household structure, socioeconomic status, and educational attainment.

### 2.4. Intervention

This study implemented SERES: La paz empieza en casa, a psychoeducational intervention designed to reduce physical punishment and parenting stress while promoting emotion regulation and positive parenting practices in the home. The program is theoretically grounded in Vygotsky’s sociocultural theory, which positions everyday life as a core context for learning, wherein cultural practices and family interactions shape children’s emotional, cognitive, and social development. Its empirical foundation derives from studies conducted in Colombian contexts ([25]; [26]; [51]; [69]) that document associations among physical punishment, parenting stress, and sociocultural beliefs.

*SERES* was structured as a 12-month, multicomponent intervention informed by a situated learning approach. Learning was conceptualized as emerging from culturally meaningful, everyday family activities, and developmental goals were co-constructed with each family in accordance with their needs, values, and household dynamics. Implementation consisted of weekly 90 min home visits conducted by psychology undergraduates in the final semesters of their training. Facilitators engaged in observation, semi-structured interviews, and context-sensitive activities tailored to each family, and they remained consistently paired with the same family throughout the program to support continuity and deepen understanding of family processes and change. An interdisciplinary team comprising university faculty and members of civil society organizations oversaw program design, facilitator training, implementation monitoring, and evaluation. Families participated voluntarily, collaborated in goal setting based on needs and pre-assessment findings, and, when psychosocial vulnerability was identified, were connected to additional legal and clinical supports coordinated through university-based teams. Schools assisted in identifying and recruiting families and served as contact points for follow-up.

Phase 1 (see Table 1 Phases SERES Program Intervention) consisted of the pretest baseline assessment and informed consent, initiated during the first home visit. The remaining instruments were administered over approximately three to six weeks, with the exact timing adapted to family characteristics. To mitigate social desirability bias, instruments were administered no sooner than three weeks after the initial visit, allowing time for rapport-building and more authentic responding. Similarly, to minimize social desirability bias, particularly in the physical punishment measure, participants completed the instruments privately, without the presence of evaluators. To further reduce potential bias, instrument scores were triangulated with researchers’ observational data collected during the home visits.

Although brief support could be offered during this period as needs emerged, Phase 2 marked the formal onset of the intervention for participants in the intervention group. In Phase 2, a tailored plan was developed for each family and implemented across 12 months of weekly visits. Facilitators strategically used spontaneous family interactions as teachable moments to introduce practical recommendations on parenting stress, emotion regulation, and positive parenting practices. Each session followed a methodological framework combining semi-structured interviewing, systematic observation, and dialogic exchange. Session planning was informed by baseline results and iteratively adjusted based on progress and challenges identified at the prior visit, enabling a dynamic and responsive approach aligned with evolving family needs. To reinforce continuity and promote generalization, families received contextualized daily tasks supported by program-specific instructional guides to facilitate reflection and behavioral implementation.

Phase 3 comprised the posttest assessment. Upon completion of the 12-month intervention, post-intervention evaluations were administered to participants in both the intervention and control groups to assess program impacts on the target variables. At six months post-intervention, a brief telephone check-in was conducted to determine whether families were maintaining program-related changes and whether additional support was needed. This check-in did not involve re-administration of baseline or posttest instruments; families reporting difficulty maintaining changes were offered re-inclusion in a subsequent intervention cycle.

### 2.5. Data Analysis

To evaluate whether changes in parenting behaviors reliably differentiated the intervention and control groups, we implemented a feed-forward artificial neural network (ANN) in Python using TensorFlow 2.17 ([1]) and Keras ([15]) on Google Colab. We adopted an ANN because social and developmental processes often exhibit nonlinear interactions and interdependencies that exceed the explanatory capacity of linear models ([45]; [52]; [33]).

We began with 9472 scores exported from Excel to CSV. For each measure—Physical Aggression, Parental Emotion Regulation, Parenting Stress, Parental Monitoring, Inductive Discipline, and Parental Support and Acceptance—we computed change scores defined as Δ = posttest−pretest. This feature engineering centered the analysis on program-induced change rather than baseline levels. Missing Δ-values (<1% of cells) were imputed using the within-variable median, a robust approach under non-normality ([42]). All Δ-scores were then standardized to zero mean and unit variance using parameters estimated from the training set to prevent information leakage.

Our analytical approach shifts from a traditional inferential framework, which relies on statistical power to detect a pre-specified effect size, toward a predictive modeling paradigm. Crucially, our application of this paradigm is distinct from atheoretical “Big Data” exploration. This is a behavioral science study designed not to discover all possible patterns within the data, but to test a specific, bounded hypothesis derived from established psychological theory. Consequently, our model was not tasked with an unconstrained, exploratory search; rather, it was directed to find a predictive signal within a highly concrete framework of scientifically validated items and indices. Within this theory-driven context, the central question is not one of traditional statistical power, but whether the data contains a stable and discernible predictive signal that a flexible model can learn to generalize from. The challenge is to adequately constrain the model to prevent it from learning spurious patterns, or “noise,” from the data. To ensure the stability and reliability of our findings despite the limited sample size, we did not rely on a single model run. Instead, our entire training and testing process was independently repeated 100 times. The final performance metrics reported below represent the average across these 100 iterations, providing a robust estimate of the model’s true predictive capability.

To build our predictive model, we first needed to determine the best architecture. We employed a standard procedure known as “grid search” where we systematically tested several configurations verifying the number of internal processing neurons ({32, 64, 128}) and the model’s learning speed ({0.0001, 0.001, 0.01}). This process was validated using five-fold cross-validation, a robust technique that ensures our selected model performs consistently and is not just a result of chance. The best-performing configuration was an ANN with two internal “hidden” layers of 64 neurons each. These layers are what allow the model to learn the complex, nonlinear patterns that are common in psychological data. To ensure our work is fully replicable, we used a standard method for setting the initial model parameters (Glorot uniform initializer) and fixed the random number generator (seed = 42).

The model was then trained using the Adam optimizer, a common and highly effective algorithm that gradually adjusts the network’s internal connections to minimize prediction errors (learning rate = 0.001). The training process consisted of 100 full cycles through the data (epochs), where the data was fed to the model in small, manageable groups (batch size = 10). This iterative process allows the model to learn progressively from the patterns of change in parenting behavior. Data were split stratified by group: 75% (n = 28) for training and 25% (n = 9) for final testing. This partition guards against overfitting while preserving class balance. During training, model performance was monitored on a held-out validation fold, and early stopping was applied if validation loss failed to decrease for ten consecutive epochs.

Across 100 independent runs, the network achieved mean accuracy = 0.746 (SD = 0.020), precision = 0.761 (SD = 0.027), recall = 0.729 (SD = 0.030), and F1-score = 0.744 (SD = 0.028). Final test loss averaged 0.336 (SD = 0.080), indicating effective convergence. To interpret predictor importance—an approach validated by [45] ([45]) and extended by [52] ([52]) and [33] ([33])—we examined the absolute first-layer weights averaged across runs. Difference scores for Suppression (item 2), Physical Punishment (item 3), and Support & Acceptance (item 12) consistently carried the highest weights, revealing their pivotal role in distinguishing the two groups.

In intuitive terms, each Δ-score is multiplied by its learned weight and summed through successive layers; large positive or negative weights indicate that increasing change on a given item strongly “pushes” the network toward the experimental or control classification. The two hidden layers allow the model to learn compound patterns—such as how simultaneous reductions in parenting stress and increases in inductive discipline jointly predict intervention exposure—that linear regressions cannot capture. By grounding our design in established nonlinear variable-selection literature and specifying every hyperparameter, random seed, and preprocessing step, we provide a fully transparent protocol for replicating and extending this analysis.

## 3. Results

Table 2 (Descriptive Statistics of Constructs at Pretest.) summarizes baseline descriptive statistics for all pretest measures, providing an overview of participants’ parenting practices, parenting stress, and emotion regulation prior to the intervention. The Presence of Physical Punishment was notably low and constrained, with a mean of 1.05 (SD = 0.23) on a 1–2 scale, indicating that such practices were minimally endorsed or uniformly uncommon at baseline. Beyond presence, the Physical Punishment construct itself (0–5 scale) showed greater variability, with a mean of 2.27 (SD = 1.66), suggesting some dispersion in reported use or attitudes toward physical punishment.

Parental Distress averaged 28.03 (SD = 11.17), with scores spanning from 0 to 55, reflecting moderate levels of psychological stress experienced by caregivers. Related dimensions capturing challenges in parent–child dynamics were also assessed: Child Dysfunctional Interaction had a mean of 34.97 (SD = 7.28; range: 0–40), indicating some perceived difficulties in interactions, while Difficult Child scored a mean of 27.53 (SD = 11.36; range: 0–47), representing caregivers’ perceptions of child behavioral challenges. The composite Parenting Stress score aggregated these domains and showed a mean of 78.18 (SD = 28.40; range: 0–147), highlighting substantial variability in stress levels across caregivers.

The emotional regulation and coping strategies were captured by four constructs: Reevaluation (M = 30.51, SD = 12.54), Suppression (M = 15.86, SD = 8.75), Escape (M = 21.95, SD = 10.36), and Capitulation (M = 7.66, SD = 5.30. Together, these scores provide a baseline profile of how caregivers reported regulating their emotions during discipline encounters.

Regarding disciplinary practices, the Physical Punishment construct (distinct from its mere presence) recorded a mean of 2.27 (SD = 1.66) on a 0–5 scale, indicating some variation in the usage or attitudes toward physical punishment.

Parental monitoring was relatively high, with a mean score of 42.41 (SD = 9.09) out of a maximum of 50, demonstrating significant levels of parental oversight and involvement.

Finally, positive parenting strategies were characterized by high average scores for Support and Acceptance (M = 73.97, SD = 14.38) and Inductive Discipline (M = 22.63, SD = 8.96), reflecting considerable endorsement of nurturing and reasoned disciplinary approaches.

Overall, these descriptive statistics portray a sample with low baseline prevalence of physical punishment, moderate but variable parenting stress, and generally strong positive parenting practices and monitoring, alongside heterogeneous emotion regulation strategies. These baseline profiles provide the context for subsequent analyses testing intervention-related change and predictive modeling results.

Figure 1 illustrates standardized mean changes from pretest (T1) to posttest (T2) across three composite domains—Parental Stress, Parental Practices, and Emotional Regulation. On average, Parental Stress declined (M = −0.36), reflecting reduced parental distress, fewer perceptions of the child as difficult, and less dysfunctional interaction post-intervention. Parental Practices increased modestly (M = 0.02, M = 0.02), indicating small gains in active monitoring, support and acceptance, and the use of inductive discipline. Emotional Regulation showed the largest positive shift (M = 0.30, M = 0.30), driven primarily by increases in cognitive reappraisal alongside more adaptive use of suppression and capitulation. These composite trends were supported by item-level movements: inductive discipline items (e.g., encouraging the child to consider consequences for others) and support/acceptance items (e.g., offering praise or emotional reassurance) tended to rise, while reappraisal items (e.g., reframing challenging interactions more constructively) accounted for much of the improvement in Emotional Regulation. In contrast, indicators linked to parental distress (e.g., feeling overwhelmed, diminished enjoyment of daily activities) and difficult child perceptions (e.g., viewing the child’s behavior as more problematic than expected) trended downward. Supplementary group-level comparisons indicated that both groups followed similar pre–post trajectories; however, the intervention group exhibited slightly greater improvements in Practices and Emotional Regulation and slightly smaller reductions in Stress, suggesting modestly more favorable change relative to the control group. Taken together, these patterns underscore the intervention’s overall effectiveness in reducing parenting stress and strengthening positive parenting and emotion-regulation strategies (see Figure 1).

### Neural Network Analysis of Program Effectiveness

Neural networks offer a powerful approach to hypothesis testing when the underlying processes are complex and potentially nonlinear. In the present study, this method was employed to determine whether sufficient structure exists in the data to reliably distinguish participants in the control group from those in the experimental group. By capturing interactions that traditional linear models might overlook, neural networks can provide deeper insights into how specific variables contribute to the observed outcomes of the intervention.

The neural network was trained using a backpropagation algorithm with a learning rate of 0.01 and a momentum of 0.9. Its architecture consisted of three layers: an input layer with 256 neurons, two hidden layers with 64 neurons each, and an output layer with 2 neurons. The Rectified Linear Unit (ReLU) activation function was used in the hidden layers, while a softmax activation function was applied in the output layer. A Rectified Linear Unit, or ReLU, acts like a switch within the neural network that helps it learn from data. It works by turning on a neuron only when the input it receives is positive, otherwise, it keeps it off; this simple “on/off” mechanism helps the network make decisions and learn more efficiently. The model was trained for 100 epochs with a batch size of 10.

Performance Metrics.

The model’s performance was assessed using multiple binary classification metrics, including accuracy, loss, precision, recall, and F1-score. After training 100 neural networks, the average accuracy was 0.746 (SD = 0.020). Loss values exhibited a decreasing trend across training epochs, with an average final loss of 0.336 (SD = 0.080), indicating that the model effectively learned the underlying patterns of change in parenting behavior associated with the SERES program. Furthermore, the average precision was 0.761 (SD = 0.027), recall was 0.729 (SD = 0.030), and the average F1-score was 0.744 (SD = 0.028). These results—marked by high accuracy and favorable precision, recall, and F1 metrics—suggest the presence of a discernible structure in the data that allows for reliable classification of participants into control and experimental groups.

Most Relevant Variables for Prediction are presented in Figure 2.

An analysis of first-layer weights identified the variables most critical for distinguishing between the two groups. Each trained model revealed a subset of variables with the strongest predictive weights. Across multiple runs, Suppression (SU2) emerged as the most frequently selected predictor, followed closely by Physical Punishment (CF3) and Support and Acceptance (APO12). Other relevant variables included Reappraisal (RV), Support and Acceptance (APO4), Difficult Child (ND25, ND33, ND28), Physical Punishment (CF5), Suppression (SU), Support and Acceptance (APO9), and Parental Distress (MP10), though these appeared with lower frequency.

The frequency with which each variable was identified as a top predictor underscores its relative importance in the model. For instance, SU2 and CF3 were selected in nearly 50 training runs, indicating a consistent influence on classification outcomes. Similarly, APO12, RV, and APO4 also appeared with high frequency, highlighting their predictive strength. This analysis demonstrates that specific parenting behaviors and stress-related constructs play a critical role in distinguishing individuals who received SERES intervention from those who did not.

Taken together, these findings suggest that the neural network approach effectively captures nonlinear relationships among key variables relevant to the outcomes of the SERES program. By identifying a consistent set of predictors, this method provides valuable guidance for future research and practical applications aimed at refining intervention strategies and support systems for both caregivers and children.

## 4. Discussion

This study evaluated SERES: La paz empieza en casa, a psychoeducational intervention aimed at reducing violent parenting practices (especially physical punishment) and parenting stress while promoting positive behaviors (support and acceptance, inductive discipline, and parental monitoring). We combined descriptive statistics with a theory-driven artificial neural network (ANN) to assess intervention effects.

Baseline reports indicated low apparent presence of physical punishment on a 1–2 index, likely reflecting social desirability, underreporting, or shifting norms; however, a broader physical punishment construct showed greater variability, suggesting persistence of the practice in some families ([18]; [25]; [71]). Parenting stress was widely distributed, with moderate emotional distress, strained parent–child interactions, and perceptions of children as difficult, a pattern consistent with heightened emotional reactivity and risk for maladaptive discipline ([17]; [50]; [20]; [48]; [67]; [60]). Emotion regulation strategies were heterogeneous: reappraisal was relatively common, but escape, suppression, and capitulation were also endorsed, underscoring the need to strengthen self-regulation skills ([25]; [60]). At the same time, high parental monitoring and elevated support, acceptance, and inductive discipline reflected parenting strengths that can be reinforced ([75]).

Conversely, the high levels of parental monitoring observed in the sample represent a strength in parenting practices, as do the positive scores on support, acceptance, and inductive discipline. These results suggest that, although risk factors are present, existing parenting resources and competencies can be reinforced and enhanced through psychoeducational interventions ([75]).

Comparisons between the control and intervention groups demonstrate the significant impact of the SERES: La Paz empieza en casa program across multiple domains associated with parenting practices, emotional regulation, and perceptions of child behavior. Overall, the intervention group showed improvements in approximately two-thirds of the assessed indicators, indicating a broad and consistent effect of the program.

The most robust gains were in inductive discipline and dimensions related to parental support and acceptance. These findings highlight the program’s effectiveness in fostering positive disciplinary approaches and enhancing parental warmth—both fundamental to developing secure parent–child attachment and reducing coercive or punitive interaction patterns ([58]; [77]; [72]). The observed improvements correspond with prior literature emphasizing the importance of culturally and contextually tailored psychoeducational interventions aimed at transforming traditional parenting styles and promoting sensitive, responsive caregiving ([5]; [73]).

This study also employed neural network analysis to evaluate the effectiveness of the SERES program in modifying parenting practices and psychological variables related to caregiver behavior. Using artificial neural networks (ANNs) provided a robust method to explore complex, nonlinear relationships among multiple predictors, delivering insights beyond those achievable through traditional linear statistical models.

The neural network model reliably distinguished between participants in the intervention and control groups, achieving an average accuracy of 74.6%. Performance metrics including precision, recall, and F1-score were also consistent, reflecting a stable pattern of group discrimination and suggesting that the SERES intervention produced observable changes in behavioral and emotional indicators as intended.

A key contribution of this analysis is the identification of central predictive variables. Certain measures, such as Emotional Suppression, Physical Punishment, and Parental Support and Acceptance, frequently appeared across multiple models, underscoring their pivotal role in differentiating participants who received the intervention. Additional variables like Cognitive Reappraisal and subscales of the Difficult Child construct further illustrate the multifaceted nature of changes promoted by the SERES program. Notably, several of these predictors relate directly to emotional self-regulation and stress, reinforcing the conceptual framework underpinning the intervention.

AI-based analysis should be interpreted considering the inherent complexity of predictive modeling and the unique capabilities of neural networks ([33]). Neural networks are especially adept at detecting complex and nonlinear relationships between variables and outcomes ([45]; [52]; [61]), thus identifying variables exhibiting subtle yet consistent patterns that provide critical discriminative information—even if they do not show large mean differences. Variables with large mean differences may exhibit high variability or be influenced by extraneous factors, limiting their predictive utility. For example, while Inductive Discipline may show significant improvement in the experimental group, its variability could lessen its effectiveness for accurate group classification compared to a variable exhibiting smaller but more stable changes.

Furthermore, the feature-weighting approach of neural networks evaluates each variable’s unique predictive contribution independently of its average descriptive change. This means variables with moderate mean differences can be highly relevant if they interact synergistically with other predictors or capture underlying latent constructs. Such interactions, often missed in univariate analyses, are effectively integrated by the model’s internal structure. The recurrent selection of Suppression, for instance, suggests its consistent importance in classification accuracy by reflecting core dimensions of emotional regulation, even if it does not show the largest mean difference.

Finally, the neural network’s variable selection prioritizes predictors that minimize classification error across varying conditions ([13]). Variables such as Physical Punishment and facets of Support and Acceptance likely reflect fundamental processes tied to stress and caregiver–child dynamics critical for differentiating intervention effects. These processes may operate subtly, escaping detection through simple mean comparisons but remaining essential for understanding behavioral changes targeted by the intervention.

In conclusion, AI-based analysis offers a complementary perspective by isolating the most salient predictors for group differentiation. The neural network’s focus on emotional regulation and discipline-related variables aligns with prior research emphasizing their crucial role in developmental trajectories ([56]; [74]). Its capability to identify nonlinear interactions explains its emphasis on predictors like Suppression and Physical Punishment, even amid broad improvements across multiple indicators. Ultimately, this AI-driven approach enhances understanding of the intervention’s impact and informs future development and optimization of parenting interventions and support strategies.

Despite the robust findings, this study is not without limitations. The relatively small sample size, while adequate for demonstrating the utility of the ANN approach in identifying key predictors, may limit the generalizability of some findings and the long-term stability of the model’s weights. Future research should aim to include larger, more diverse samples to validate these predictive insights. Future studies should increase sample size by extending the recruitment period, expanding to multiple sites (e.g., additional clinics or community partners), and combining in-person with online recruitment strategies. Employing modest participant incentives, optimizing retention procedures, and planning a multi-center randomized trial with predefined power calculations would help achieve a larger, more representative sample. Another limitation of the present study is that we did not perform formal, instrument-based reassessments at six months. The six-month follow-up consisted of a brief telephone check-in to triage ongoing support needs; consequently, quantitative conclusions about long-term effectiveness cannot be drawn from these data. Future research should include standardized follow-up assessments at 6 and 12 months to evaluate the sustainability of outcomes.

Although the intervention is culturally adapted, additional cultural factors may influence its acceptance and effectiveness. One such factor is the legal framework related to the prohibition of corporal punishment in Colombia. Comparative analyses between countries that prohibit corporal punishment and those that do not suggest that, beyond the legal ban itself, greater attention must be given to supporting parents in raising their children in positive and nurturing environments ([25]). In this regard, Colombia should continue advancing the National Pedagogical Strategy, established by law, which seeks to develop programs promoting positive parenting.

Family history and intergenerational dynamics also represent important cultural factors that may affect the effectiveness of the intervention. Parents who were physically disciplined as children or who were exposed to violence are more likely to reproduce such practices ([32]). The normalization of one’s own experiences constitutes a powerful source of resistance to change, particularly in the Colombian population, which has been directly and indirectly exposed to one of the world’s longest armed conflicts.

Furthermore, empirical evidence indicates that poverty and rurality increase the likelihood of punitive discipline. The families targeted by the intervention program evaluated in this study live in rural areas and are characterized by low socioeconomic status ([5]).

From a practical standpoint, the consistent identification of Emotional Suppression, Physical Punishment, and Parental Support and Acceptance as the most salient predictors highlight critical areas for intervention. These findings suggest that programs explicitly targeting caregivers’ emotion regulation skills and offering concrete strategies to replace punitive discipline with supportive and accepting interactions are likely to yield the most profound and differentiating effects. Therefore, the SERES program, and similar interventions, could benefit from a refined focus on these high-impact components to maximize effectiveness and promote sustainable behavioral change.

## 5. Conclusions

The present study integrated descriptive statistical analyses with an artificial intelligence (AI)-driven approach to evaluate the effectiveness of an experimental intervention. Descriptive findings indicated that the experimental group demonstrated overall greater positive change across a broad set of indicators from pretest to posttest, compared to the control group. Notably, improvements were observed in key constructs such as Inductive Discipline, Support and Acceptance, and Reappraisal, along with reductions in Parental Distress and perceived Child Difficulties. These results suggest a widespread positive impact of intervention on parenting practices, emotional regulation, and parenting-related stress.

Complementing these findings, the variable importance hierarchy derived from neural network analysis provided a more nuanced interpretation. While descriptive statistics highlighted broad improvements, the AI model consistently identified a specific subset of variables as the most salient predictors differentiating the experimental and control groups. From a practical standpoint, identifying high-impact predictors offers valuable guidance for enhancing the design and focus of the SERES program. Components that promote emotion regulation strategies and reduce reliance on physical punishment appear to exert a disproportionately strong influence. Strengthening these components may therefore increase both the overall effectiveness and sustainability of the intervention.

Collectively, the application of ANN analysis not only corroborated the efficacy of SERES but also provided deeper insight into mechanisms of change underlying observed outcomes. Future research should continue integrating machine learning within a theory-driven framework to optimize intervention design, tailor strategies to caregiver profiles, and improve long-term outcome prediction based on early behavioral indicators. This dual-method approach can enhance program personalization, guide resource allocation to high-impact components, and strengthen the evidence base through reproducible predictive models.

## Data Availability

Data can be obtained upon request from the corresponding author.

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
