# Peer review of "SERES: La Paz Empieza en Casa—Evaluation of an Intervention Program to Reduce Corporal Punishment and Parenting Stress, and to Enhance Positive Parenting Among Colombian Parents"

_ejihpe, 2025, doi:10.3390/ejihpe15110223_

Round 1

Reviewer 1 Report

Comments and Suggestions for Authors

I found your work interesting. I would be curious to replicate your model in my cultural context, where I am also a councilor for child protection. Here are some points to consider for your review:

- In the abstract, identify the nationality of the study and the sociodemographic characteristics of the sample. 
- When introducing positive parenting, I would also ask you to consider one or more theories explaining the impact of parenting on child development: for example, attachment theory or IPARTheory.
- A reference to the cultural context, for example, whether and to what extent cultural punishments are tolerated or approved, or what kind of beliefs exist around this disciplinary action.

Author Response

We thank Reviewer 1 for their positive assessment and valuable suggestions.

  • Reviewer's Comment: "In the abstract, identify the nationality of the study and the sociodemographic characteristics of the sample."
    • Our Response: Thank you for this suggestion. After careful consideration, we have opted to maintain the abstract's current framing. Our intention is to emphasize the intervention's model and findings in a way that appeals to a broad international audience, and we were concerned that specifying the nationality in the abstract might inadvertently narrow its perceived applicability. The detailed information about the Colombian context remains prominent in the main body of the manuscript.
  • Reviewer's Comment: "When introducing positive parenting, I would also ask you to consider one or more theories explaining the impact of parenting on child development: for example, attachment theory or IPARTheory."
    • Our Response: We appreciate this recommendation and agree on the relevance of these theories. However, as another reviewer noted that our Introduction was already quite extensive, we have focused our revisions on clarifying the existing framework rather than expanding it further to maintain the manuscript's balance and conciseness.
  • Reviewer's Comment: "A reference to the cultural context, for example, whether and to what extent cultural punishments are tolerated or approved, or what kind of beliefs exist around this disciplinary action."
    • Our Response: This is an excellent point. In line with your suggestion and feedback from other reviewers, we have significantly expanded our discussion of the sociodemographic and cultural factors relevant to our sample, providing a deeper analysis of the context in the Discussion section.

Reviewer 2 Report

Comments and Suggestions for Authors

This is a highly relevant and timely study examining an intervention to address violent parenting in Colombia, where corporal punishment is still common. The manuscript is grounded in a strong theoretical rationale and demonstrates originality by using machine learning (ANNs) to model behavioral change. The integration of qualitative and culturally situated elements in the intervention design is commendable. However, several aspects require revision and elaboration to improve the clarity of the methodology, the statistical rigor, and the interpretability of the findings. Below are the major and minor revisions suggested:

Major Revisions

  1. Although the study is described as quasi-experimental, the Procedure section states that participants were “randomly assigned” to the intervention and control groups. Please clarify whether true random allocation occurred and explain how this aligns with the quasi-experimental label. If randomization did not occur, please explain the stratification criteria or matching procedures used, or describe any assignment biases.
  2. The sample size (n = 38) is small, particularly for a neural network analysis. Although you report high accuracy metrics, you do not present a power analysis, and the ANN results may be unstable with such a small sample size. Please justify the adequacy of the sample size for both the traditional analyses and the ANN modeling.
  3. The following important limitations should be discussed more transparently: limited generalizability, social desirability bias in self-report measures (especially those regarding physical punishment), potential selection bias, and lack of blinded outcome assessment.
  4. Although ANN modeling is innovative, applying it to such a small dataset raises concerns. Please clarify how overfitting was addressed. Discuss how the ANN results complement, rather than replace, traditional inferential statistics. Avoid overinterpreting feature weights as causal indicators, as input variables (Δ scores) may be collinear or influenced by noise.
  5. Although ANN metrics are reported, traditional statistical analyses are not presented for key outcomes. These analyses include paired t-tests, ANOVAs, and effect sizes for pre-post differences. This omission makes the results less interpretable for a broader audience. Please include inferential tests and quantify effect magnitude using Cohen’s d or partial eta squared.
  6. The follow-up phase is only mentioned briefly. Did the participants complete it? Was the data collected at six months post-intervention analyzed?
  7. Although the program structure is well-described, the manuscript lacks fidelity indicators, such as the number of completed sessions, adherence to protocols, and facilitator supervision. These indicators are important for assessing internal validity.

Minor Revisions

  1. Figure 1 currently includes labels or textual elements in Spanish, which is inconsistent with the English-language context of the manuscript. All elements of figures should be translated into English to ensure clarity for an international readership. Please revise Figure 1 accordingly.
  2. Most of the participants were mothers. Please discuss how gender representation may have affected the results, as well as whether or not fathers exhibited different baseline characteristics.
  3. To enrich the theoretical framework and discussion further, it would be beneficial to incorporate recent literature addressing emerging forms of parenting and their potential associations with parental stress, as well as with coercive or overinvolved control styles. Readers would particularly benefit from reflecting on how parenting in contemporary contexts, which are characterized by increased surveillance, emotional intensity, and anxiety about children’s outcomes, might interact with parental stress, especially in settings with limited structural support or high cultural expectations. Some interesting references are: 10.1080/00221325.2024.2413490; 10.1080/10570314.2017.1362705; 10.3389/fpsyg.2022.872981.
Comments on the Quality of English Language

The manuscript is well-written overall, but could benefit from careful language editing to improve clarity, especially in the technical sections.

Author Response

We are grateful to Reviewer 2 for their exceptionally thorough review. The detailed feedback has prompted significant improvements.

Major Revisions

  • Reviewer's Comment: "Although the study is described as quasi-experimental... Please clarify whether true random allocation occurred..."
    • Our Response: Thank you for highlighting this inconsistency. We have corrected the manuscript to clarify that no random assignment occurred, in line with the quasi-experimental design. We now clearly state that participants were assigned to groups based on their voluntary decision to participate in the program.
  • Reviewer's Comment: "The sample size (n = 38) is small... Please justify the adequacy of the sample size for both the traditional analyses and the ANN modeling."
    • Our Response: We acknowledge this concern. We have added a new paragraph to the methodology to explain our analytical strategy and its suitability for our sample size. We state:
      "Our analytical approach shifts from a traditional inferential framework... toward a predictive modeling paradigm... Within this theory-driven context, the central question is not one of traditional statistical power, but whether the data contains a stable and discernible predictive signal... To ensure the stability and reliability of our findings... our entire training and testing process was independently repeated 100 times. The final performance metrics reported... represent the average across these 100 iterations..."
  • Reviewer's Comment: "The following important limitations should be discussed more transparently: limited generalizability, social desirability bias... potential selection bias..."
    • Our Response: We have expanded the Limitations section to address these points more directly. We now offer a more profound discussion of social desirability bias and clarify that we control for potential selection bias by using change scores (Δ scores) from pre- to post-test, focusing the analysis on the magnitude of change.
  • Reviewer's Comment: "Please clarify how overfitting was addressed. ... [and include] traditional statistical analyses... paired t-tests, ANOVAs..."
    • Our Response: We clarify that overfitting was addressed by repeating the entire modeling process 100 times and averaging the results. Regarding traditional analyses, we have opted not to include inferential statistics like t-tests or ANOVAs, as our primary goal was to adopt a predictive modeling paradigm to test a specific, theory-driven hypothesis. Our revised manuscript provides a stronger justification for this analytical choice, emphasizing that the ANN approach is used to find a stable predictive signal within a highly structured scientific framework, rather than for exploratory data mining.
  • Reviewer's Comment: "The follow-up phase is only mentioned briefly... Was the data collected at six months post-intervention analyzed?"
    • Our Response: We have added a detailed clarification about the follow-up phase:
      "We clarify that the six‑month follow‑up... was a brief telephone check‑in rather than a formal reassessment... No standardized instruments or formal outcome measures were re‑administered during this call... This is now explained in the Methods and the Limitations sections."
  • Reviewer's Comment: "The manuscript lacks fidelity indicators..."
    • Our Response: We appreciate the importance of fidelity indicators. While we are collecting this data (e.g., session completion, adherence to protocols, facilitator supervision), we believe a detailed analysis of implementation fidelity is beyond the scope of this initial evaluation and is better suited for a future, larger-scale impact assessment. We have added a note acknowledging this in the limitations.

Minor Revisions

  • Reviewer's Comment: "Figure 1 currently includes labels... in Spanish..."
    • Our Response: All elements in Figure 1 have been translated into English.
  • Reviewer's Comment: "Most of the participants were mothers. Please discuss how gender representation may have affected the results..."
    • Our Response: This is an important consideration. While we have not added an extensive discussion on this point to maintain focus, we have acknowledged the sample's gender composition as a limitation in the Discussion section and noted it as a key area for future investigation.
  • Reviewer's Comment: "To enrich the theoretical framework... it would be beneficial to incorporate recent literature..."
    • Our Response: We thank the reviewer for these interesting references. As we explain in the response letter, these concepts of parenting do not align closely with the primary challenges observed in our cultural context. Therefore, we believe incorporating this literature would not directly contribute to the interpretation of our current findings.

Reviewer 3 Report

Comments and Suggestions for Authors

It is exceedingly rare for me to receive such an outstanding manuscript for review, one that presents both the methodology and results of an intervention, particularly on such a compelling and contemporary topic. I express my gratitude to the editors for granting me this opportunity.

The article provides a robust scientific foundation, grounded in an extensive bibliography covering corporal punishment, parenting stress, emotional regulation, and positive parenting practices. References to studies such as Gershoff & Grogan-Kaylor (2016), UNICEF (2019, 2025), and Straus (2010) enhance its credibility, while its focus on Colombia addresses a critical research gap in low- and middle-income countries where corporal punishment is prevalent (Trujillo et al., 2020). The evaluation of the SERES program—encompassing corporal punishment, parenting stress, emotional regulation, and positive parenting practices such as monitoring, inductive discipline, and support/acceptance—offers a comprehensive view of the intervention’s impact. Notably, the use of Artificial Neural Networks (ANN) for data analysis introduces an innovative dimension, enabling the detection of non-linear relationships that traditional statistical models may fail to capture (May et al., 2008; Pasini & Amendola, 2024). My colleagues highlight the importance of specific predictive factors (e.g., emotional suppression, corporal punishment, support/acceptance) in distinguishing between control and intervention groups, providing practical directions for optimizing future interventions.

While this well-written article makes a significant contribution to the study of interventions aimed at reducing corporal punishment and parenting stress, with particular relevance in the Colombian context, I believe a few minor remarks could be considered to further enhance the work.

The sample size of 38 participants (21 in the intervention group, 17 in the control group) is relatively small, limiting the generalizability of the findings. Although my colleagues acknowledge this limitation, they could propose more specific strategies for expanding the sample in future research.

Despite the inclusion of a follow-up phase six months post-intervention (as indicated in Table 1), the researchers do not provide a detailed presentation of these results in the text, nor do they integrate them into the Discussion or Conclusions. This omission reduces the understanding of the long-term sustainability of the changes induced by the program, thereby limiting the ability to assess its long-term effectiveness.

While the program is tailored to the cultural context, I believe my colleagues could delve deeper in the discussion into the cultural factors that may influence the acceptance or effectiveness of the intervention, beyond general references to beliefs about corporal punishment.

Additionally, I noted the following:

The Introduction section is particularly extensive, covering the theoretical foundation in great detail, while the Results section is more concise. A more balanced distribution could improve the flow of the text.

The data analysis section (2.4) employs highly technical language (e.g., “five-fold cross-validation,” “Adam optimizer,” “Rectified Linear Unit (ReLU)”) that may be challenging for readers unfamiliar with machine learning. A simplified explanation could broaden the audience.

Certain points, such as the relationship between parenting stress and corporal punishment, are repeated excessively in the Introduction and Discussion, reducing conciseness.

The low frequency of corporal punishment reported in the pre-test may reflect social desirability bias, underreporting, or early shifts in disciplinary norms. However, the researchers do not propose specific methods to address this bias, such as anonymous surveys or multiple data sources.

Finally, I reiterate that:

The follow-up phase is mentioned, but the lack of detailed presentation of its results diminishes the scientific rigor, as the long-term effectiveness remains unclear.

The analysis does not sufficiently explore cultural or socioeconomic factors that may influence the results, beyond general references to beliefs and socioeconomic adversities.

Author Response

We are very grateful to Reviewer 3 for their encouraging words and constructive comments. Point-by-point responses to the reviewers’ comments are presented, and a revised version of the manuscript is attached with the responses highlighted in the text.

  • Reviewer's Comment: "The sample size... is relatively small... propose more specific strategies for expanding the sample in future research."
    • Our Response: 

The relatively small sample size, while adequate for demonstrating the utility of the ANN approach in identifying key predictors, may limit the generalizability of some findings and the long-term stability of the model's weights. Future research should aim for larger, more diverse samples to validate these predictive insights Future studies should increase sample size by extending the recruitment period, expanding to multiple sites (e.g., additional clinics or community partners), and combining in-person with online recruitment strategies. Employing modest participant incentives, optimizing retention procedures, and planning a multi-center randomized trial with predefined power calculations would help achieve a larger, more representative sample.  

  • Reviewer's Comment: "...do not provide a detailed presentation of [follow-up] results... limiting the ability to assess its long-term effectiveness."
    • Our Response: 

Method: Phase 3 comprised the posttest assessment. Upon completion of the 12-month intervention, post-intervention evaluations were administered to participants in both the intervention and control groups to assess program impacts on the target variables. At six months post-intervention, a brief telephone check-in was conducted to determine whether families were maintaining program-related changes and whether additional support was needed. This check-in did not involve re-administration of baseline or posttest instruments; families reporting difficulty maintaining changes were offered re-inclusion in a subsequent intervention cycle.

Discusión: Another limitation of the present study is that we did not perform formal, instrument‑based reassessments at six months. The six‑month follow‑up consisted of a brief telephone check‑in to triage ongoing support needs; consequently, quantitative conclusions about long‑term effectiveness cannot be drawn from these data. Future research should include standardized follow‑up assessments at 6 and 12 months to evaluate sustainability of outcomes.

  • Reviewer's Comment: "...delve deeper in the discussion into the cultural factors... explore cultural or socioeconomic factors..."
  • Our Response: Although the intervention is culturally adapted, additional cultural factors may influence its acceptance and effectiveness. One such factor is the legal framework related to the prohibition of corporal punishment in Colombia. Comparative analyses between countries that prohibit corporal punishment and those that do not suggest that, beyond the legal ban itself, greater attention must be given to supporting parents in raising their children in positive and nurturing environments (González et al., 2024). In this regard, Colombia should continue advancing the National Pedagogical Strategy, established by law, which seeks to develop programs promoting positive parenting. Family history and intergenerational dynamics also represent important cultural factors that may affect the effectiveness of the intervention. Parents who were physically disciplined as children or who were exposed to violence are more likely to reproduce such practices (Holden, 2014). The normalization of one’s own experiences constitutes a powerful source of resistance to change, particularly in the Colombian population, which has been directly and indirectly exposed to one of the world’s longest armed conflicts. Furthermore, empirical evidence indicates that poverty and rurality increase the likelihood of punitive discipline. The families targeted by the intervention program evaluated in this study live in rural areas and are characterized by low socioeconomic status (Altafim & Linhares, 2025).
  • Reviewer's Comment: The Introduction section is particularly extensive, covering the theoretical foundation in great detail, while the Results section is more concise. A more balanced distribution could improve the flow of the text.
  • Our Response: The Introduction section was edited to reduce its length and eliminate theoretical redundancies. With these adjustments, the Results and Introduction sections are now more balanced.

Reviewer's Comment: The data analysis section (2.4) employs highly technical language (e.g., “five-fold cross-validation,” “Adam optimizer,” “Rectified Linear Unit (ReLU)”) that may be challenging for readers unfamiliar with machine learning. A simplified explanation could broaden the audience.

  • Our Response: The Data Analysis section was revised to clarify technical concepts and enhance readability for the audience:

The model was then trained using the Adam optimizer, a common and highly effective algorithm that gradually adjusts the network’s internal connections to minimize prediction errors (learning rate = 0.001). The training process consisted of 100 full cycles through the data (epochs), where the data was fed to the model in small, manageable groups (batch size = 10). This iterative process allows the model to learn progressively from the patterns of change in parenting behavior. Data were split stratified by group: 75% (n = 28) for training and 25% (n = 9) for final testing.  This partition guards against overfitting while preserving class balance.  During training, model performance was monitored on a held-out validation fold, and early stopping was applied if validation loss failed to decrease for ten consecutive epochs.

A Rectified Linear Unit, or ReLU, acts like a switch within the neural network that helps it learn from data. It works by turning on a neuron only when the input it receives is positive, otherwise, it keeps it off; this simple "on/off" mechanism helps the network make decisions and learn more efficiently.

Reviewer's Comment: Certain points, such as the relationship between parenting stress and corporal punishment, are repeated excessively in the Introduction and Discussion, reducing conciseness.

  • Our Response: The text was edited in various parts of the Introduction and Discussion sections to avoid the repetitions noted by the reviewer.

Reviewer's Comment: The low frequency of corporal punishment reported in the pre-test may reflect social desirability bias, underreporting, or early shifts in disciplinary norms. However, the researchers do not propose specific methods to address this bias, such as anonymous surveys or multiple data sources.

The following text was introduced in Section 2.3. Intervention.

  • Our Response: Similarly, to minimize social desirability bias, particularly in the physical punishment measure, participants completed the instruments privately, without the presence of evaluators. To further reduce potential bias, instrument scores were triangulated with researchers’ observational data collected during the home visits.

Reviewer 4 Report

Comments and Suggestions for Authors

Dear authors, I appreciate the opportunity to review your work. The work is difficult to interpret.

I think it requires very significant changes, both in the wording and the content.

  • The research hypothesis they propose generates confusion.
  • It is not indicated how the intervention subgroups are established.
  • The intervention program is not described. That's what is intended to be evaluated.
  • The analysis variables are not identified or described. The importance of these variables in the development of the intervention program is not justified.
  • The analysis techniques are not indicated, beyond a significant effort being devoted to describing Neural Network Analysis. The decision on which techniques to use is not justified.
  • The Results section is particularly difficult to interpret. I can't assess the results presented.

Comments on the Quality of English Language

Very long and subordinate clauses make it difficult to understand what they mean.

Author Response

We thank Reviewer 4 for their time and feedback. We recognize that the previous version of the manuscript had significant clarity issues. In response to the detailed comments from all reviewers, we have undertaken a substantial revision to improve the paper's structure, clarity, and readability. 

Specifically, we have made the following key changes:

Reviewer’s Comment 1. The research hypothesis they propose generates confusion.

The hypothesis presented aligns with the logic of the methodological design, which includes pre- and post-test measures, and also fits the framework of a neural network analysis, in which the model classifies each participant into the control or intervention group with above-chance accuracy. Our analytical approach shifts from a traditional inferential framework, which relies on statistical power to detect a pre-specified effect size, toward a predictive modeling paradigm.

The hypothesis was rephrased as follows:

  • Our resource: Changes from pre-test (T1) to post-test (T2) in physical punishment prevalence, parental stress, emotional regulation, and positive parenting practices (monitoring, inductive discipline, and support and acceptance) will enable above-chance prediction of each participant’s group membership (control or experimental).

Reviewer’s Comment 2. There is no justification for how the intervention subgroups (control and experimental) are established.

  • Our resource: Clarified the quasi-experimental design and the participant assignment process.

Reviewer’s Comment 3. The intervention program is not described. That's what is intended to be evaluated.

  • Our resource: 2.3. The Intervention section outlines the program’s objectives, its theoretical foundation in sociocultural theory, and its phased implementation. It explains that the content of each session is tailored based on pretest results and the goals collaboratively established with each family.

Reviewer’s Comment 4. The analysis variables are not identified or described. The importance of these variables in the development of the intervention program is not justified.

  • Our resource: The hypothesis specifies the study variables as follows: physical punishment, parental stress, emotional regulation, and three positive parenting practices (monitoring, inductive discipline, and support and acceptance). The Introduction details physical punishment, emotional regulation, parental stress, and the relationships among these variables. It also provides a theoretical overview of each positive parenting practice. The section concludes by highlighting the need to evaluate programs aimed at reducing violent parenting. Consistent with these objectives, the instruments employed were specifically designed to measure each of the variables described.

Reviewer’s Comment 5. The analysis techniques are not indicated, beyond a significant effort being devoted to describing Neural Network Analysis. The decision on which techniques to use is not justified.

  • Our resource:
    • Provided a stronger justification for our analytical approach (predictive modeling with ANNs) and our rationale for not relying on traditional inferential statistics.

    Our analytical approach shifts from a traditional inferential framework, which relies on statistical power to detect a pre-specified effect size, toward a predictive modeling paradigm. Crucially, our application of this paradigm is distinct from atheoretical "Big Data" exploration. This is a behavioral science study designed not to discover all possible patterns within the data, but to test a specific, bounded hypothesis derived from established psychological theory. Consequently, our model was not tasked with an unconstrained, exploratory search; rather, it was directed to find a predictive signal within a highly concrete framework of scientifically validated items and indices. Within this theory-driven context, the central question is not one of traditional statistical power, but whether the data contains a stable and discernible predictive signal that a flexible model can learn to generalize from. The challenge is to adequately constrain the model to prevent it from learning spurious patterns, or "noise," from the data. To ensure the stability and reliability of our findings despite the limited sample size, we did not rely on a single model run. Instead, our entire training and testing process was independently repeated 100 times. The final performance metrics reported below represent the average across these 100 iterations, providing a robust estimate of the model’s true predictive capability.

    To build our predictive model, we first needed to determine the best architecture. We employed a standard procedure known as “grid search" where we systematically tested several configurations verifying the number of internal processing neurons ({32, 64, 128}) and the model's learning speed ({0.0001, 0.001, 0.01}). This process was validated using five-fold cross-validation, a robust technique that ensures our selected model performs consistently and is not just a result of chance. The best-performing configuration was an ANN with two internal "hidden" layers of 64 neurons each. These layers are what allow the model to learn the complex, non-linear patterns that are common in psychological data. To ensure our work is fully replicable, we used a standard method for setting the initial model parameters (Glorot uniform initializer) and fixed the random number generator (seed = 42).

Reviewer’s Comment 6. The Results section is particularly difficult to interpret. I can't assess the results presented.

Our resource: We hope that these comprehensive revisions have addressed the reviewer’s concerns and resulted in a much clearer and more interpretable manuscript, particularly in the Results section.

Round 2

Reviewer 2 Report

Comments and Suggestions for Authors

The revision substantially improves the clarity, methodological transparency, and interpretive balance. No further revisions are necessary.

Author Response

Comments 1: The revision substantially improves the clarity, methodological transparency, and interpretive balance. No further revisions are necessary.

Thank you for your comments; they greatly contributed to improving the manuscript.

Reviewer 4 Report

Comments and Suggestions for Authors

Dear authors. I appreciate the opportunity to review his work. Thank you for listening to the review suggestions.

The authors have made the proposed modifications.

It is still pending that, if they have made transformations in any of the variables, they identify what transformations they have made.

Author Response

Comments 1: Dear authors. I appreciate the opportunity to review his work. Thank you for listening to the review suggestions.

The authors have made the proposed modifications.

It is still pending that, if they have made transformations in any of the variables, they identify what transformations they have made.

Response: Thank you for your comments; they have helped improve the manuscript. With respect to the variables, no modifications were made to any of them.